# Protecting Salt Vulnerable Areas Using an Enhanced Roadside Drainage System (ERDS)

**Sepideh E. Tabrizi [1], Jessica Pringle [1], Zahra Moosavi [1], Arman Amouzadeh [1], Hani Farghaly [1], William R. Trenouth [2] and Bahram Gharabaghi [1,*]**

[1] School of Engineering, University of Guelph, Guelph, ON N1G 2W1, Canada
[2] AECOM, 250 York Street, Suite 410, London, ON N6A 6K2, Canada
* Correspondence: bgharaba@uoguelph.ca; Tel.:+1-(519)-824-4120 (ext: 58451)

**Abstract:** De-icing road salt application as a part of winter road maintenance is a standard practice with over 60 billion kilograms applied to roads worldwide each winter to ensure traffic safety. However, high concentrations of chlorides in melted ice and snow runoff from roads and parking lots can have adverse effects on both surface and ground water, especially in salt vulnerable areas. A salt vulnerable area is a sensitive area to road salts where additional salt management measures may be required to mitigate potential adverse environmental effects. The main objectives of this paper were to present a new design method for sizing Enhanced Roadside Drainage Systems (ERDS), demonstrate the findings of a 3 year field monitoring and to assess the long term performance of the ERDS design using PCSWMM. A new conceptual design of ERDS was also modelled to demonstrate its effectiveness in protecting salt vulnerable areas. To showcase the new design method, we completed two case studies, one for a relatively pristine headwater stream and one for a moderately impacted urban stream. Stormwater management models were developed for the two scenarios—with and without the ERDS—to assess the benefits of the new system and its effectiveness in protecting salt vulnerable areas at each site.

**Keywords:** road runoff quality; water quality; salt vulnerable areas; groundwater protection

## 1. Introduction

Road traffic safety in cold climates is a challenging issue for road authorities [1–6]. The application of rock salt (mainly sodium chloride) and brine dispensation are de-icing and snow-melting techniques used to maintain road safety during winter conditions—typically from late November to April. Population growth and urbanization are the leading causes of steadily increasing rates of annual total salt use on roads and parking lots—today, over 60 billion kilograms of road salt is applied to roads worldwide [7–9].

Highway runoff from snow and ice melt is conveyed to receiving streams through surface drainage systems such as swales and roadside ditches [10]. Despite the importance of road safety in winter, road salt application has negative impacts on receiving streams and can result in acute chloride toxicological effects on aquatic life [11–14]. A number of studies have investigated the impacts of de-icing agents on salt vulnerable areas, as well as new methods for reducing chloride concentrations in urban streams [7,15–22]. Roadside drainage systems can be designed to capture polluted highway runoff, improve water quality, attenuate peak flows, and mitigate shock loadings of chlorides in receiving streams [21]. The concentration of chlorides in meltwater infiltrating into the ground also threatens drinking water supplies and aquatic species [14,23].

Environment Canada published a Code of Practice related to best management practices for the Environmental Management of Road Salts in 2004, under the Canadian Environmental Protection Act (1999), to manage and reduce the use of salt. In Ontario, the Stormwater Management Planning and Design Manual [24], the Low Impact Development

Stormwater Management Planning and Design Guide [25], and the Low Impact Development Stormwater Management Guidance Manual [26] have been published to protect water resources and the environment. Enhanced Roadside Drainage System (ERDS) is evolving as an alternative to traditional approaches in stormwater management. Therefore, this study aims to develop methodology for optimized ERDS design as a critical tool for curtailing stormwater management costs while protecting surface and groundwater quality in cold climate regions of the world.

## 2. Methodology

### 2.1. Influence of Road Salt on Aquatic Life

The Canadian Council of the Ministers of Environment (CCME) [15] provides guidelines on the short-term and the long-term chloride toxicity levels for aquatic life. The CCME has derived the log-normal distribution curves of chloride concentration shown in Figure 1 from the short-term 50% Effective and Lethal Concentration values (LC/EC50s), and long-term no and low species impacted to chloride toxicity in freshwater.

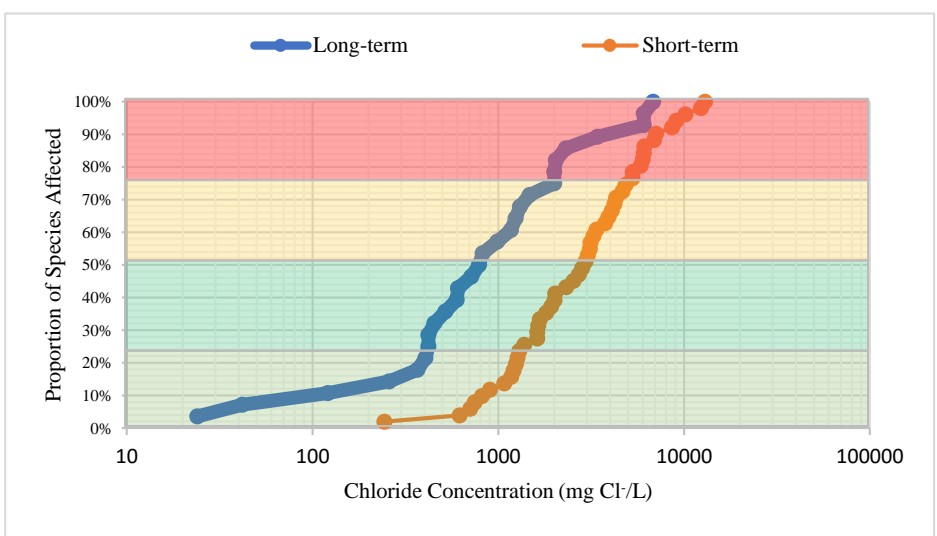

**Figure 1.** Chloride concentration for species exposed to chloride in freshwater [15].

The Event Mean Concentration (EMC) approach is the common method for environmental impact characterization [27]. In this method, the flow weighted concentration is calculated for each storm event [28].

### 2.2. Field Site Monitoring

ERDS practices utilize modern design techniques to manage stormwater through a combination of treatment controls and runoff reduction. Bioretention cells in ditch areas are a type of ERDS feature that consist of a vegetated surface layer, a soil filtration layer and a granular storage layer. The surface water from highway runoff is infiltrated into the soil layer, and it is detained in the storage layer. The storage layer then slowly releases the receiving water (in a controlled fashion) by considering hydraulic conductivity and porosity of granular media [25,29].

For the purposes of this study, a bioretention cell with a membrane liner was constructed on the northbound side of Highway 412, near Whitby, Ontario (Figure 2). The surface runoff from a roughly 1485 m$^2$ catchment area discharges from the northbound lanes of Highway 412 to the ERDS (Figure 2b).

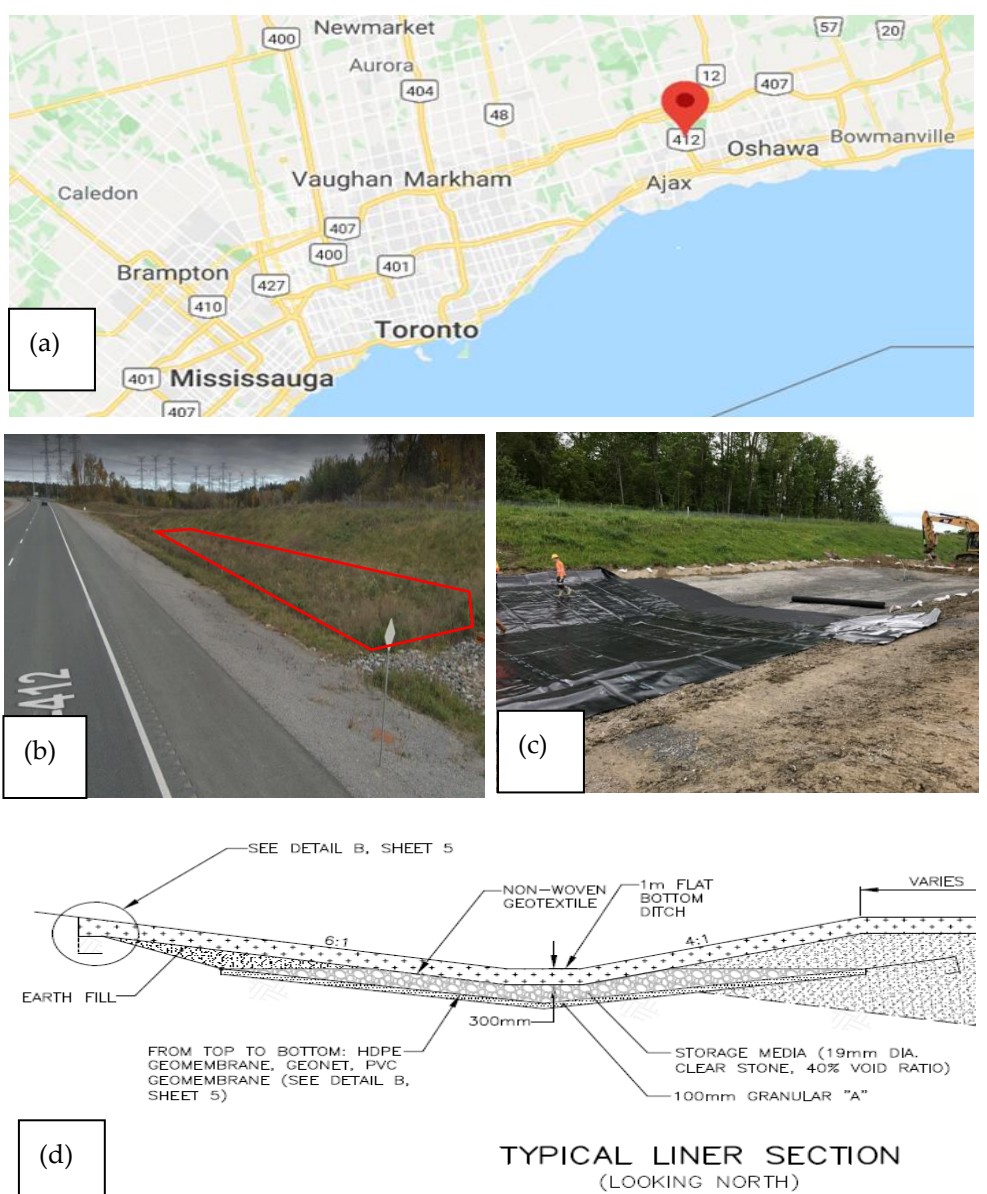

**Figure 2.** (**a**) Study site location; (**b**) Roadside ditch; (**c**) Photo of the ERDS during the installation; (**d**) Cross-section of the ERDS.

In 2019, an electrical conductivity sensor (INW CT2X, manufactured by INW/Seametrics Kent, Washington, WA, USA) with complimentary water level logger was installed in the monitoring chamber downstream of the ERDS to measure the flow and water quality discharged from the highway to the receiving stream. Observed chloride concentration was calculated based on measured electric conductivity (*EC*) using Equation (1) [28].

$$Cl = 375.49(EC) - 296.12 \tag{1}$$

where *Cl* = chloride concentration (mg/L) and *EC* = electrical conductivity (mS/cm).

Figure 3 shows the relationship between rainfall and runoff volumes for monitored storm events at the Highway 412 site. The imperviousness or runoff coefficient in the catchment regression (mostly highway pavement surface plus a portion of the gravel shoulder) is 0.98, and the initial abstraction is approximately 5 mm.

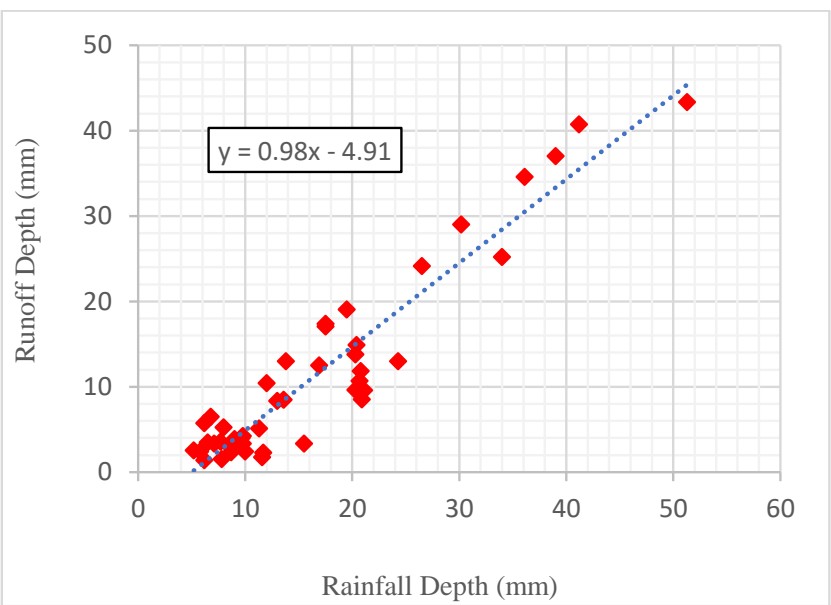

**Figure 3.** Mass balance of rainfall and runoff volumes for storms.

### 2.3. ERDS Model for Two Case Studies

To assess the function and benefits of the ERDS within the context of this research study, the Personal Computer Storm Water Management Model (PCSWMM) was used to numerically model the observed performance of the constructed pilot system, in addition to the behaviour and performance of a typical roadside ditch. PCSWMM is a spatial decision support system for stormwater management modelling (SWMM) that was developed by Computational Hydraulics International (CHI). PCSWMM uses a GIS engine that was built upon the upgraded U.S. EPA SWMM5 engine to be able to model types of low impact development (LID) features such as bioretention cells and infiltration trenches [30]. This paper presents a novel ERDS design methodology for two case studies using PCSWMM modelling, which allows for water quality mass balance analysis between areas upstream and downstream of the ERDS in a receiving stream.

Due in large part to its environmental impacts, road deicing salt specifically has be analyzed in the current work. Due to the crucial role deicing salts play in improving road safety, its use is difficult to eliminate entirely. However, appropriate stormwater management techniques and suitable best management practices are recommended for implementation in salt vulnerable areas [31]. Trenouth [21] presented the ERDS design framework that can be used to protect sensitive areas. Detention storage volumes and system outlet orifice diameters are key optimization parameters in ERDS design for improving environmental performance.

In addition to the optimization technique undertaken in this research a novel conceptual ERDS design was investigated in the second case study. The new design places the ERDS under the shoulder of the higway which allows a simpler and potentially more economic design to be be used. The new design allows for a rectangular cross section that does not need to follow the shape of the roadside ditch and therefore, can use much less liner material. The cross section of the conceptual design is shown in Figure 4, the ERDS design can be integrated within a typical permeable shoulder. The new ERDS was modelled as being on either side of Highway 407 and 401. The modelling was done for a ten-year period.

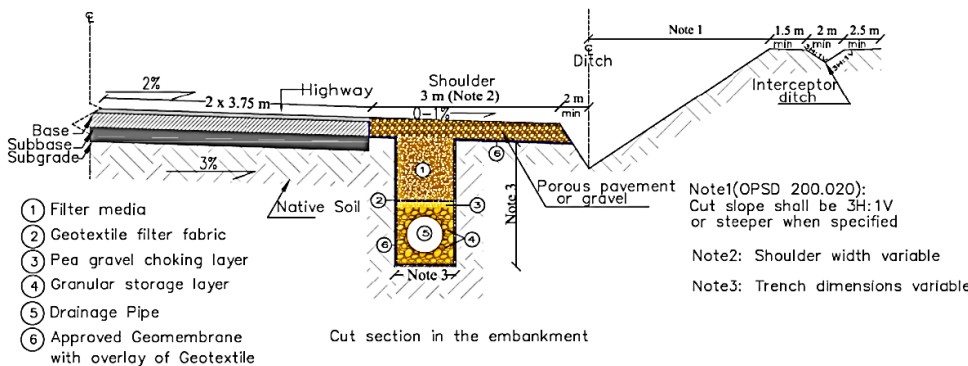

**Figure 4.** The ERDS design on the shoulder of Highway 407 and 401.

In this novel ERDS design the minimum depth and width of the storage and the suitable outlet orifice diameter using PCSWMM model. The first step in the novel design technique was to consider a very large storage volume with a large underdrain orifice size. The ERDS storage and orifice size were then gradually reduced until minimal species were exposed to their L/EC50 and L/EC10 chloride concentration limits, as defined by the CCME guidelines.

The controlled highway effluent flow rate to the receiver is provided through the sizing of a circular orifice according to Darcy's law [31]. The ERDS volume consists of a perforated pipe and a granular media designed using a granular media type O with a mean porosity (*n*) of 30%, and a mean hydraulic conductivity of the 10th percentile granular size using Hazen's method [32]. To avoid upwelling, a seepage discharge rate in the ERDS plus a perforated pipe discharge rate are sized to be less than the orifice peak flow.

First, a case study was simulated for a pristine headwater stream. This case study includes two segments of Highway 412 within the Lynde Creek watershed, which were assessed and simulated during the field component of this research. Figure 5 presents the delineated catchment areas draining to the ERDS. For each segment, one of these cells has been installed within the northbound ditch, and the other installed within the southbound ditch of Highway 412. Total length and width of each lane has been estimated to be 1700 m and 3.75 m, respectively. The flow monitoring data from Water Survey of Canada hydrometric stations 02HC018 (100 km$^2$ drainage area) and 02HC054 (39 km$^2$ drainage area) within the Lynde Creek watershed were used to calibrate the PCSWMM model for the watershed. Hourly precipitation data was obtained from Environment Canada weather station (ID 6155875) located near the watershed (43°55′22.000″ N, 78°53′00.041″ W). In this study, the benefits of the ERDS on water quality improvements were assessed using PCSWMM model simulation outputs, comparing chloride concentrations in the receiving stream at the upstream versus the downstream of the highway both with and without ERDS scenarios.

The second case study was simulated for a moderately impacted stream to study the ERDS's performance by modelling the roadside ditch servicing the eastbound and westbound lanes of Highways 407 and 401 in southern Ontario, within the Credit River watershed (Figure 6). In this case, study, the Don River West Branch chloride concentration and flow data from Water Survey of Canada hydrometric station 02HC005 (88.1 km$^2$ drainage area) was applied in the receiving stream upstream of Highway 407 outfall to assess the combined impacts of Highways 407 and 401, downstream of Highway 401 for, with and without ERDS scenarios. Hourly precipitation data was applied from the same station as the first case study.

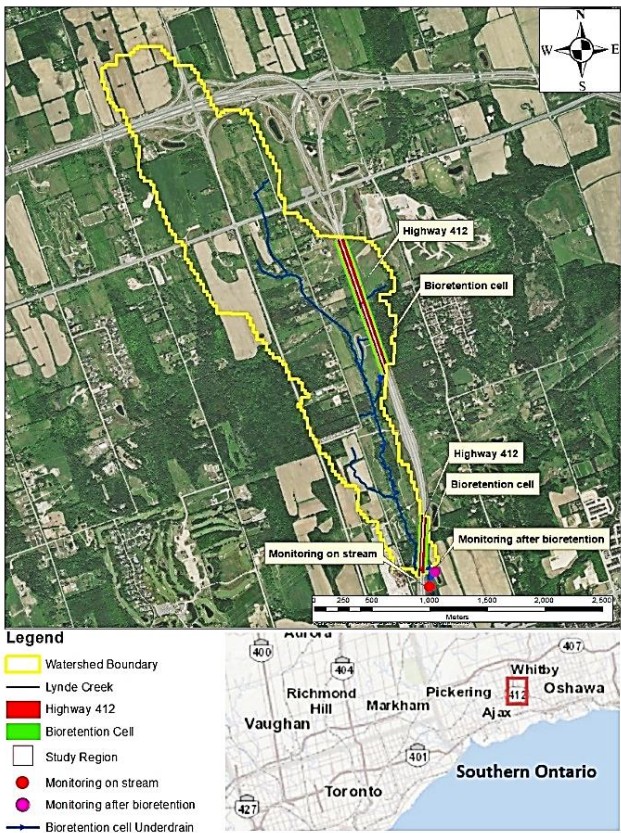

**Figure 5.** ERDS locations within the Lynde Creek watershed.

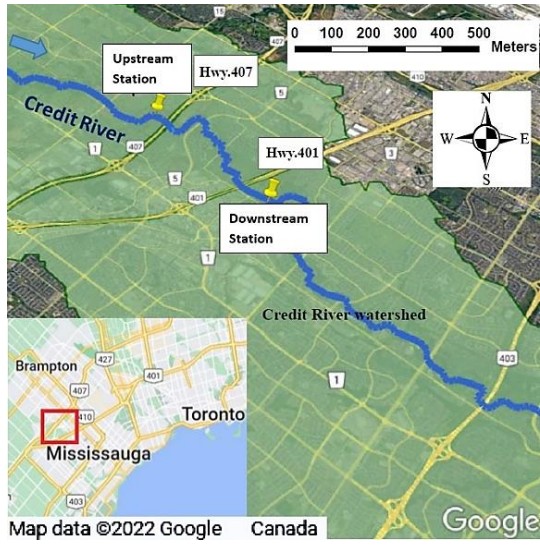

**Figure 6.** Case study site on Highways 407 and 401 at Credit River.

### 2.4. Calibration of PCSWMM for the First Case Study

PCSWMM includes water quality auto-calibration, the ability to define and compare different scenarios at the same time, and ability to determine the sensitivity of each parameter using the Sensitivity-based Radio Tuning Calibration (SRTC) tool as part of the calibration and validation process. The field site monitoring data in conjunction with the SRTC tool was used to calibrate the PCSWMM model from 2019 to 2022 on Highway 412 within the Lynde Creek watershed. The observed versus simulated flow rate and chloride concentration for three winter events are presented in Figure 7. The results demonstrated

that the simulated flow rate and chloride concentration were calibrated according to the observed field data while making a reasonable mass balance for both water and salt.

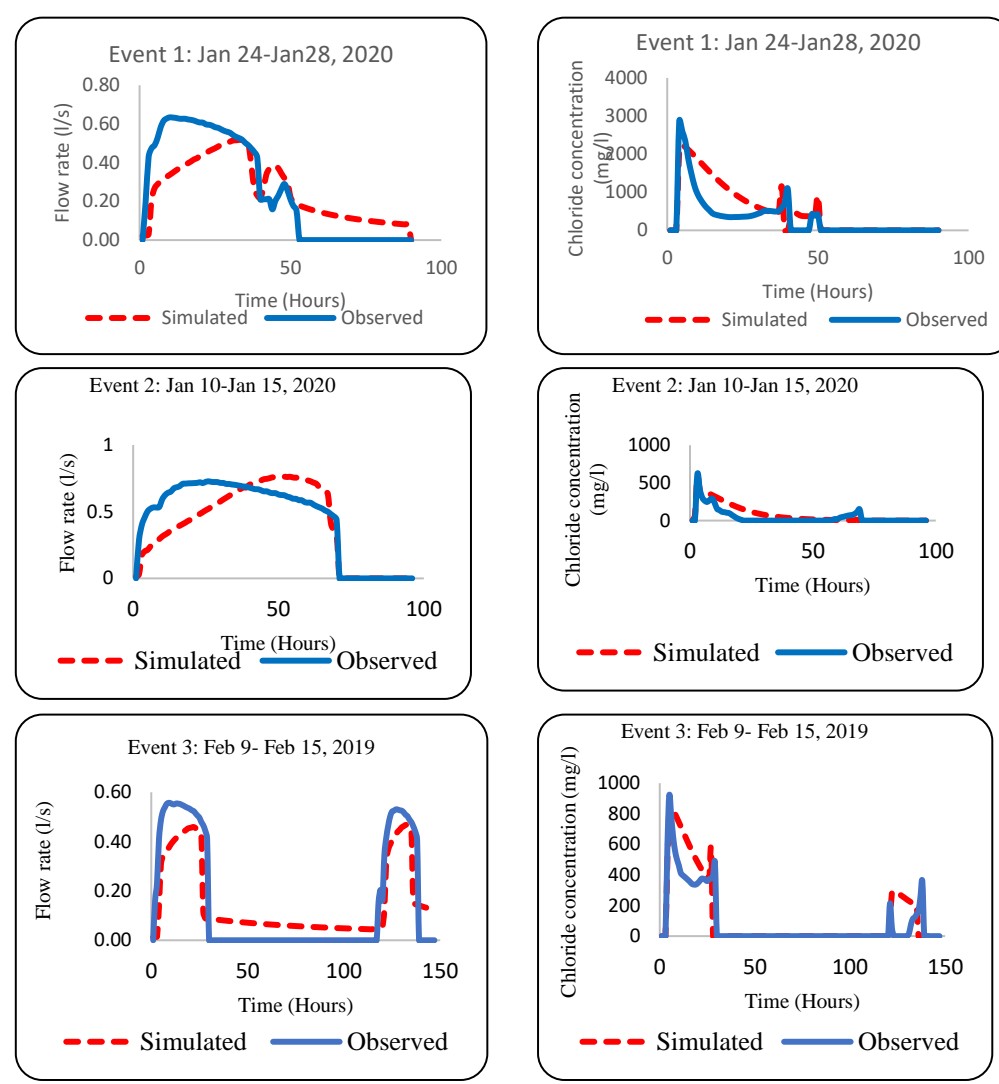

**Figure 7.** Flow rate and chloride concentration in the case study site on Highway 412.

The performance of the simulated PCSWMM model was evaluated in Table 1 against a set of coefficient of determination ($R^2$), Root Mean Square Error (RMSE), Mean Absolute Error (MAE), and Percent BIAS (PBIAS) for three winter events.

**Table 1.** Statistical parameters used to evaluate the performance of PCSWMM model.

| Indices | Coefficient | Event 1 24–28 January 2020 | Event 2 10–15 January 2020 | Event 3 9–15 February 2019 |
|---|---|---|---|---|
| Flow rate | $R^2$ (%) | 72.70 | 74.61 | 83.0 |
| | RMSE (L/s) | 0.15 | 0.16 | 0.11 |
| | MAE (L/s) | 0.13 | 0.12 | 0.09 |
| | PBIAS (%) | 4 | 12 | 1 |
| Chloride concentration | $R^2$ (%) | 65.31 | 73.38 | 74.22 |
| | RMSE (mg/L) | 432.73 | 71.5 | 122.48 |
| | MAE (mg/L) | 260.1 | 48.92 | 59.08 |
| | PBIAS (%) | 36 | 43 | 27 |

The calibrated PCSWMM model showed the lowest errors for simulation of flow rate, with $R^2$ of 83%, RMSE of 0.11 L/s, MAE of 0.09 L/s and PBIAS of 1%. The $R^2$, RMSE, MAE and PBIAS for simulated chloride concentration were calculated to be 74.22%, 122.48 mg/L, 59.08 mg/L and 27%, respectively, for Event 3 at the Highway 412 site.

## 3. Results and Discussion

### 3.1. ERDS Perfoemance for Pristine Headwater Streams

The objective of the case study of Highway 412 pilot ERDS was to assess the long term performance of the facility using the calibrated PCSWWM to determine and confirm the adequacy of the design. The facility was designed to mitigate the adverse impacts of roads salt, and it is based on the concept of capture and controlled release of highway runoff, resulting in the attenuation of peak chloride concentrations in the receiving stream (Lynde Creek). Figure 8 presents the ERDS performance data, which demonstrates how this system significantly attenuates the peak highway runoff discharge to the Lynde Creek to minimize the chloride shock to the aquatic life.

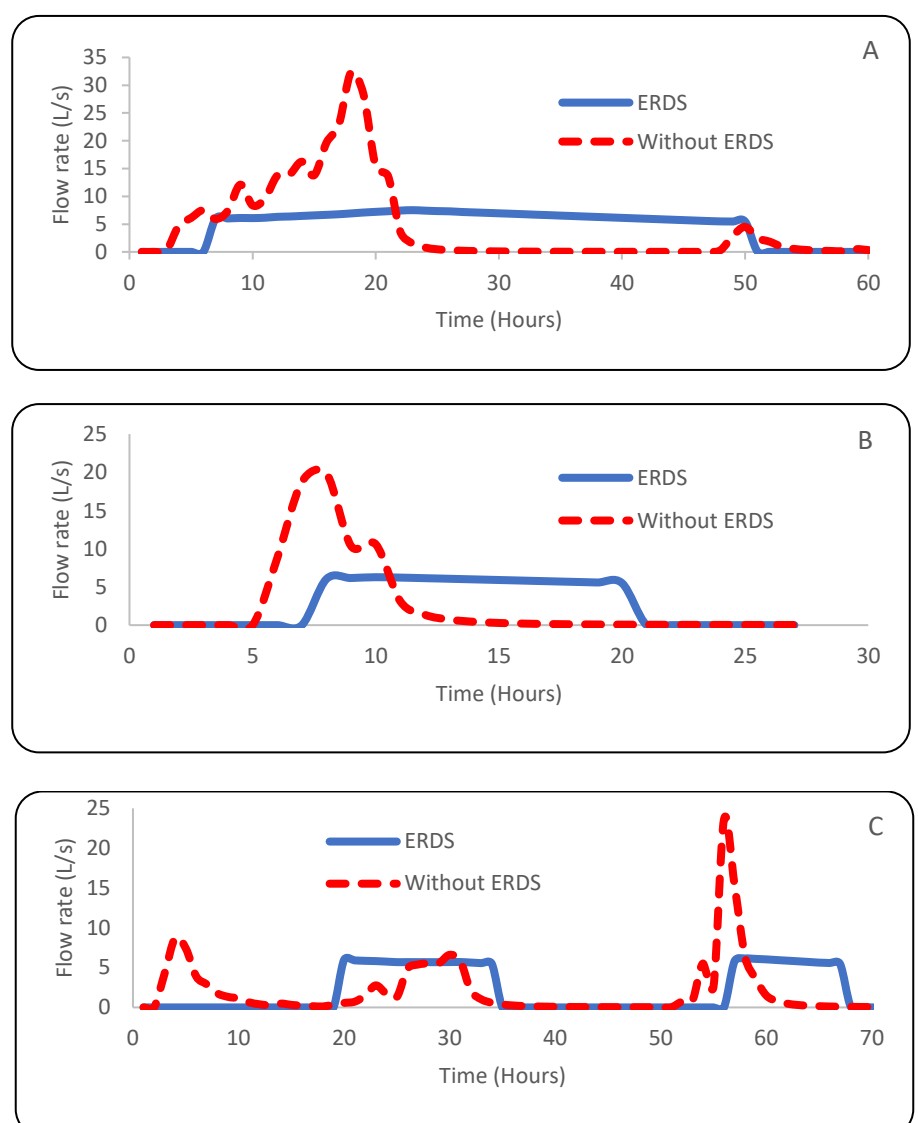

**Figure 8.** The modeled peak flow reduction for events occurring on (**A**) 24 January 2020. (**B**) 18 February 2020 (**C**) 1 to 4 March 2020 for two scenarios during the winter season.

Mean annual and standard deviation of chloride concentration in the Lynde Creek upstream of the Highway 412 are less than 60 mg/L and 5 mg/L, respectively, according to data obtained from the Ontario's Provincial Stream Water Quality Monitoring Network station (ID 06010800202). The Lynde Creek upstream of the Highway 412 crossing is a pristine headwater stream with very low chloride concentrations that would fall under Zone 1 classification (see Figure 1—less than 5% of aquatic species impacted). Therefore, we designed an ERDS that would ensure the downstream chloride concentrations would also meet the CCME acute and chronic exposure limits for this zone.

Figure 9 presents the average daily chloride concentration over a ten-year simulation period of winter seasons, and the CCME acute exposure guideline has been plotted for reference. Additionally, Figure 10 presents the average monthly chloride concentrations in Lynde Creek downstream of Hwy 412 with and without the ERDS system, along with the chronic chloride exposure limit of 120 mg/L, which is to be adhered to in order to ensure the protection of sensitive aquatic life that may be present in more pristine headwater streams.

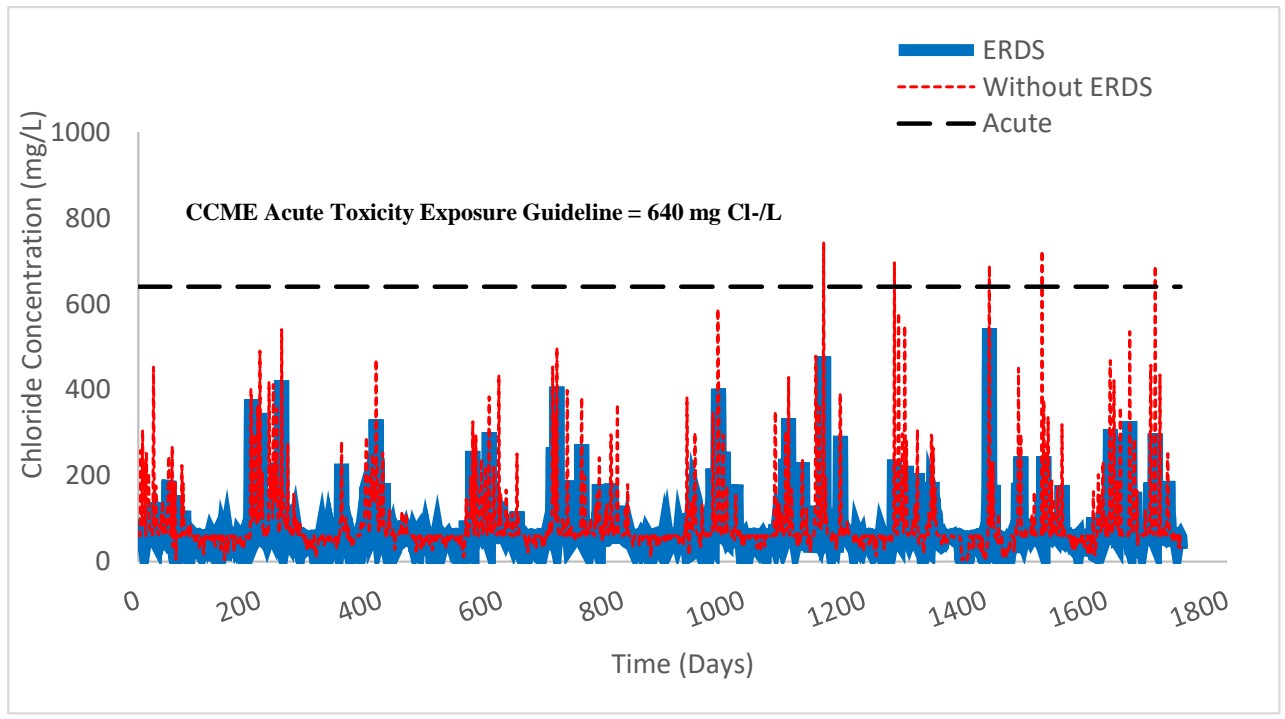

**Figure 9.** Average daily chloride concentration in the receiving stream with and without ERDS for ten winter seasons.

*3.2. ERDS design for Moderately Impacted Streams*

Runoff from various sizes of highways which are separated into east side and west side of the river with approximately same lengths, is drained into the river within the Credit River watershed. A reasonable storage volume and suitable orifice diameter were summarized in Table 2 based on different number of lanes on each side of a highway having a 3.75 m width of each lane, and different lengths of a highway catchment on each side of the river within the Credit River watershed. The length and width of the Highways were considered to be 5 km on each side of the river according to the length of highways crossing from the watershed boundary to the river, for 5 lanes of Highway 407 and 12 lanes of Highway 401, on each direction the of Highways.

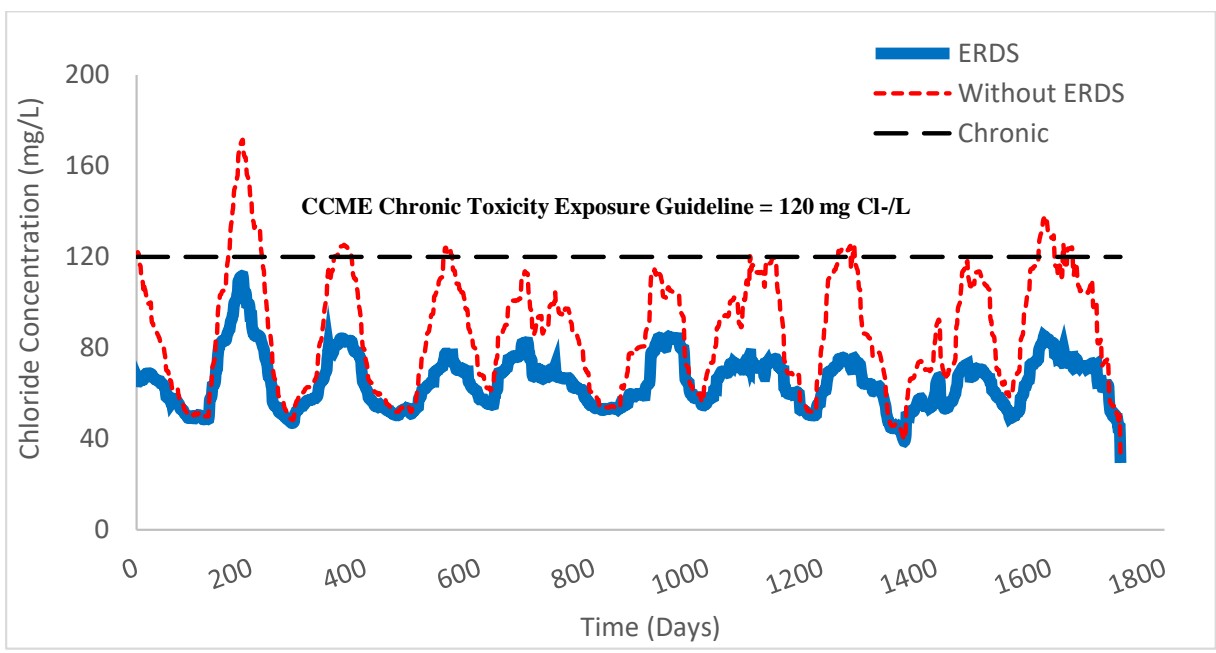

**Figure 10.** Average monthly chloride concentration in the receiving stream with and without ERDS for ten winter seasons.

**Table 2.** Summary of ERDS size and orifice diameter for each scenario.

|  | Highway Length | ERDS Size | Orifice Diameter |
|---|---|---|---|
| 2 Lanes | 2 km | 1 m width, 1 m depth | 2 Inches (0.0508 m) |
| 2 Lanes | 4 km | 1 m width, 1 m depth | 2 Inches (0.0508 m) |
| 2 Lanes | 5 km | 1 m width, 1 m depth | 2 Inches (0.0508 m) |
| 5 Lanes | 2 km | 1 m width, 1 m depth | 2 Inches (0.0508 m) |
| 5 Lanes | 4 km | 1 m width, 1 m depth | 2 Inches (0.0508 m) |
| 5 Lanes | 5 km | 1 m width, 1 m depth | 2 Inches (0.0508 m) |
| 12 Lanes | 2 km | 1 m width, 1 m depth | 2 Inches (0.0508 m) |
| 12 Lanes | 4 km | 1 m width, 1 m depth | 3 Inches (0.0762 m) |
| 12 Lanes | 5 km | 1 m width, 1 m depth | 3 Inches (0.0762 m) |

The event mean chloride concentrations were plotted against duration of exposure and four different zones were developed (Figure 11). These zones represent the quality of the stream to serve as suitable habitat for different species of aquatic life (the percentage of aquatic species that would not survive, based on their respective tolerance to chloride concentrations exposure). Figure 11 shows that the ERDS can help improve water quality in the receiving stream, such that water quality downstream of the highway is in the same water quality impact zone as the stream water quality upstream of the highway.

The results presented in Figure 11 demonstrate that with the new ERDS design methodology, the cluster of the event mean concentrations falls within the same zone of impact upstream and downstream of the highway, and that the number of species impacted are similar. This indicates that with the ERDS, performs well and creates a situation in which little to no adverse effect—due to aquatic exposure to highway runoff—would be expected to occur. Conversely, without the ERDS, conditions result in a larger number of species being at risk due to the elevated instream chloride concentrations.

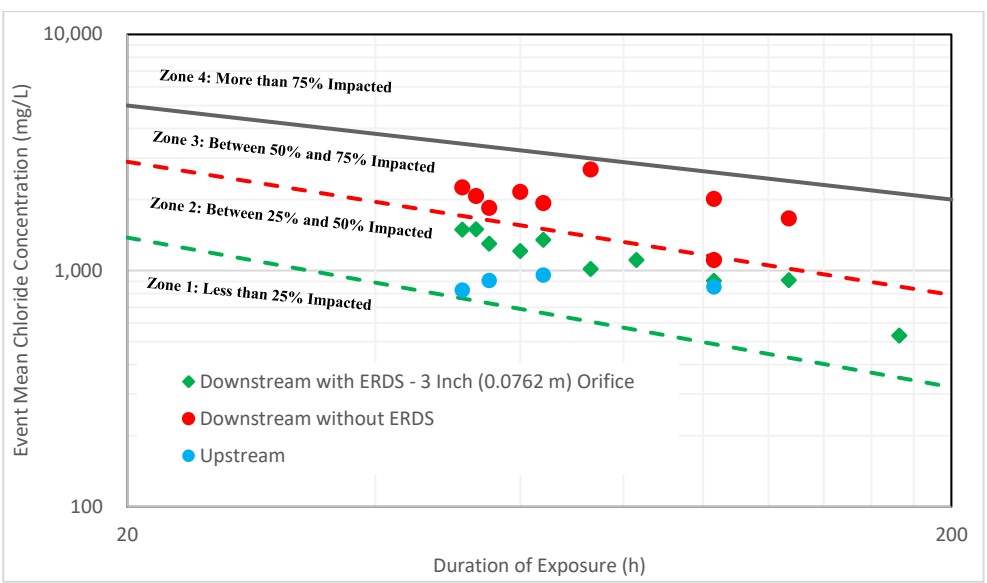

**Figure 11.** Event mean chloride concentration in the Credit River upstream and downstream of the Highways 407 and 401 with and without the ERDS.

## 4. Conclusions

This paper summarizes three years of field monitoring of a pilot study for Enhanced Road Drainage System (ERDS) constructed on Highway 412 and modelling to assess the efficacy of the system in reducing chloride shock loads and improving water quality in urban streams in salt vulnerable areas. The calibrated SWMM model was employed to simulate the ERDS capture and controlled release of salt-laden highway runoff for two scenarios: one with and one without the ERDS application to show the benefits of the system.

We presented a new design guidance for ERDS for a highway crossing over pristine headwaters (Highway 412 crossing over Lynde Creek) as well as a moderately impacted urban stream (Highway 401 crossings over Credit River). The results show that the ERDS provides appreciable improvement in water quality in the winter months when road salt is applied as part of routine winter road maintenance. Freshwater ecosystems are threatened when chloride concentrations exceed the defined acute and chronic chloride toxicity thresholds, and the ERDS design presented help to mitigate such exceedances.

The ERDS design serves as an effective mitigation system for enhancing water quality and improving the state of stormwater management. This design method provides detention and gradual release of chloride-laden runoff to downstream areas.

A new conceptual design was modelled and shows great promise to provide a potentially cost effective water quality enhancement solution even in moderately impacted stream. The concept of designing an ERDS to maintaining the upstream to downstream mean daily concentrations was examined and shows great promise in reducing the impact of highway salt applications in salt vulnerable areas.

**Author Contributions:** Conceptualization, S.E.T., H.F. and B.G.; data set preparation, S.E.T. and B.G.; methodology, S.E.T., H.F., A.A. and B.G.; modelling, S.E.T., A.A., Z.M. and B.G.; resources, S.E.T., H.F. and B.G.; writing—original draft preparation, S.E.T., H.F., A.A., W.R.T. and B.G.; writing review and editing, S.E.T., J.P., W.R.T., H.F. and B.G.; visualization, S.E.T.; supervision, H.F. and B.G.; funding acquisition, B.G. All authors have read and agreed to the published version of the manuscript.

**Funding:** The authors would like to thank the Ministry of Transportation of Ontario (MTO) Grant number 050235, and the Natural Sciences and Engineering Research Council of Canada (NSERC) Grant number 400675, for their generous funding as well as Environment and Climate Change Canada (ECCC) for the data sets.

**Institutional Review Board Statement:** Not applicable.

**Informed Consent Statement:** Not applicable.

**Data Availability Statement:** Not applicable.

**Conflicts of Interest:** The authors declare no conflict of interest.

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
