# Peer review of "Protecting Salt Vulnerable Areas Using an Enhanced Roadside Drainage System (ERDS)"

_water, doi:10.3390/w14223773_

Round 1

Reviewer 1 Report

The paper by Tabrizi et al. "Protecting Salt vulnerable Areas Using an Enhanced Roadside Drainage System (ERDS), ID: water-2000913, describes a very interesting research based on a 3-years field monitoring pilot study to assess the efficacy of ERDSs in reducing chlorine pollution in salt vulnerable areas. The results evidence the positive effect of this technology, and the future scenarios defined by PCSWM (Personal Computer Storm Water Management) model under conditions with or without ERDS are enlightening. The methodology and technology devised for this pilot study might inspire similar research in other vulnerable areas all over the world. I strongly advise the publication of this paper on Water.

I have only very few minor suggestions for the improvement of the paper:

1. Line 71: if possible provide a definition and short description of bioretention cells (what they are made of, how they work, etc..);

2. Fig 2d: provide a higher resolution picture of the ERDS cross-section, in its present form it is hardly readable on printed paper;

3. Provide the definition of the PCSWMM acronym the first time it is cited in the manuscript (not abstract), I think it is line 94:

4. Line 159: R2 with 2 as upper script;

5. Lines 212-215: please, can you check the numbers of the reported zones as it appears to me that they don't agree with what reported in Figure 10. As I understand it, zone 2 is comprised between the green and red dashed lines (therefore green and blue dots are clustered in zone 2 not zone 3), and zone 3 is comprised between the red dashed line and the grey continuous line (therefore red dots are clustered in zone 3 not zone 4). Am I right? If not, can you provide a new and clearer version of Figure 10 that does not allow this kind of misunderstanding?

Author Response

Reviewer #1

  1. Line 71: if possible provide a definition and short description of bioretention cells (what they are made of, how they work, etc..);

Response: We greatly appreciate the efforts in reviewing our paper and for providing valuable suggestions. We have included a short description of bioretention cells in the manuscript (Please see lines 71-76 in the revised manuscript).

  1. Fig 2d: provide a higher resolution picture of the ERDS cross-section, in its present form it is hardly readable on printed paper;

Response: Thank you for this comment. We have recreated Figure 2 at a higher resolution for improved clarity/quality.

  1. Provide the definition of the PCSWMM acronym the first time it is cited in the manuscript (not abstract), I think it is line 94:

Response: Thank you for this comment. We have included a definition of “PCSWMM” within the revised manuscript, starting at line 98manuscript.

  1. Line 159: R2 with 2 as upper script;

Response: Thank you for this comment. We revised R2 to R2 in the manuscript.

  1. Lines 212-215: please, can you check the numbers of the reported zones as it appears to me that they don't agree with what reported in Figure 10. As I understand it, zone 2 is comprised between the green and red dashed lines (therefore green and blue dots are clustered in zone 2 not zone 3), and zone 3 is comprised between the red dashed line and the grey continuous line (therefore red dots are clustered in zone 3 not zone 4). Am I right? If not, can you provide a new and clearer version of Figure 10 that does not allow this kind of misunderstanding?

Response: Thank you for this comment. You are correct that blue and green dots clustered in Zone 2, and red dots clustered in Zone 3 as well. We have revised the text (lines 237-240) in the manuscript to reflect this.

Reviewer 2 Report

Authors propose a new design method for sizing Enhanced Roadside Drainage Systems (ERDS) and assess the performance of optimized ERDS using PCSWMM to protect the salt vulnerable areas. The new design method is tested on two case studies, a relatively pristine headwater stream and a moderately impacted urban stream.

Authors deal with an interesting and current issue that inevitably needs to be managed. The topic is well presented and the application to the case studies is also clear. However, the present reviewer asks the authors to evaluate the following points:

·        Authors introduce the advanced road drainage system (ERDS) as an alternative to traditional approaches in stormwater management, but they do not provide any bibliography on the subject. I believe that this part needs improvement

·        It is advisable to describe, albeit succinctly, the operation of the ERDS system, perhaps in correspondence with the figure that shows its cross section.

·        Similarly, I think it is appropriate to integrate some information about the device before talking about its application. For example, what criteria was used to decide its positioning? its dimensions are a function of some parameter?

·        Is the application of the ERDS sustainable from an economic-management point of view? Especially for the dimensions of the system used in the case studies here presented.

Author Response

Reviewer #2

  • Authors introduce the advanced road drainage system (ERDS) as an alternative to traditional approaches in stormwater management, but they do not provide any bibliography on the subject. I believe that this part needs improvement

Response: We are very thankful for your positive opinion and valuable suggestions about our research. We have made efforts to succinctly summarize the significance of this research (Section 1 of the text), as well as the relevant preceding work which the work summarized expands upon (Section 2 of the text).  Notwithstanding this, we have included further summary of additional literature and a short description of ERDS in section 2.3 of the manuscript (Please see lines 98-119 in the revised manuscript).

  • It is advisable to describe, albeit succinctly, the operation of the ERDS system, perhaps in correspondence with the figure that shows its cross section.

Response: Thank you for this comment. A cross section of the ERDS system has been  included in Figure 4 in section 2.3 of the manuscript. This figure is complimentary to the in-text description which has been included in Section 2.2 of the manuscript.

  • Similarly, I think it is appropriate to integrate some information about the device before talking about its application. For example, what criteria was used to decide its positioning? its dimensions are a function of some parameter?

Response: Thank you for this comment. We have expanded section 2.3 of the manuscript to address this comment, as well as the previous suggestions.

  • Is the application of the ERDS sustainable from an economic-management point of view? Especially for the dimensions of the system used in the case studies here presented.

Response: Thank you for your comment. The scope of our manuscript is limited to the optimization of the ERDS size and orifice diameter, which are critical to mitigating the adverse impacts of roads salt. While we have noted in our manuscript that such systems are generally limited to consideration for application in headwater stream area, locations where sensitive species may be present, or other sensitive areas, our intention is to discuss the issue of cost optimization in a future paper.